# KNOWLEDGE TRANSFER VIA STUDENT-TEACHER COLLABORATION

## ABSTRACT

Accompanying with the flourish development in various fields, deep neural networks, however, are still facing with the plight of high computational costs and storage. One way to compress these heavy models is knowledge transfer (KT), in which a light student network is trained through absorbing the knowledge from a powerful teacher network. In this paper, we propose a novel knowledge transfer method which employs a **S**tudent-**T**eacher **C**ollaboration (STC) network during the knowledge transfer process. This is done by connecting the front part of the student network to the back part of the teacher network as the STC network. The back part of the teacher network takes the intermediate representation from the front part of the student network as input to make the prediction. The difference between the prediction from the collaboration network and the output tensor from the teacher network is taken into account of the loss during the train process. Through back propagation, the teacher network provides guidance to the student network in a gradient signal manner. In this way, our method takes advantage of the knowledge from the entire teacher network, who instructs the student network in learning process. Through plentiful experiments, it is proved that our STC method outperforms other KT methods with conventional strategy.

## 1 INTRODUCTION

Deep neural networks have produced breakthrough results in various fields, such as computer vision (Krizhevsky et al., 2012; He et al., 2016) and natural language processing (Mikolov et al., 2010) in recent years. Through a mass of studies (Neyshabur et al., 2017; Canziani et al., 2016; Novak et al., 2018), researchers have proved that a DNN with larger capacity will have a better generalizability, which leads to a better performance. However, larger capacity will also cause heavier computational costs and storage, making these powerful models difficult to meet real-time requirements on embedded systems.

One way to compress these heavy models is knowledge transfer (KT). As can be seen in Figure 1(a), KT is a method to improve the performance of a light student network by absorbing the knowledge from a strong teacher network. In the early studies of KT such as knowledge distillation (KD) (Hinton et al., 2015), researchers took advantage of the output vector from teacher networks, converted it into "soft target" and trained the student network with the soft target and the ground-truth. KD can only be applied in classification task since the "soft target" is produced by the softmax function with temperature T. In recent studies, many methods focused on the intermediate representation of the teacher network, in which the feature map (Romero et al., 2015), attention map (Zagoruyko & Komodakis, 2016a) or the factor (Kim et al., 2018) extracted from student network are induced to mimic the corresponding one from teacher network by minimizing the difference between them. Figure 1(b) and Figure 1(c) are the overview of AT and FT. As can be seen, the role of the teacher network in these two methods is simply to provide an intermediate representation for imitation while does not give extra help during the training process. Moreover, due to divergence of the structure between the student and teacher networks, the student network usually cannot generate the same intermediate representation as the teacher network. In this case, an intermediate representation with the smallest difference does not equal to an accurate prediction.

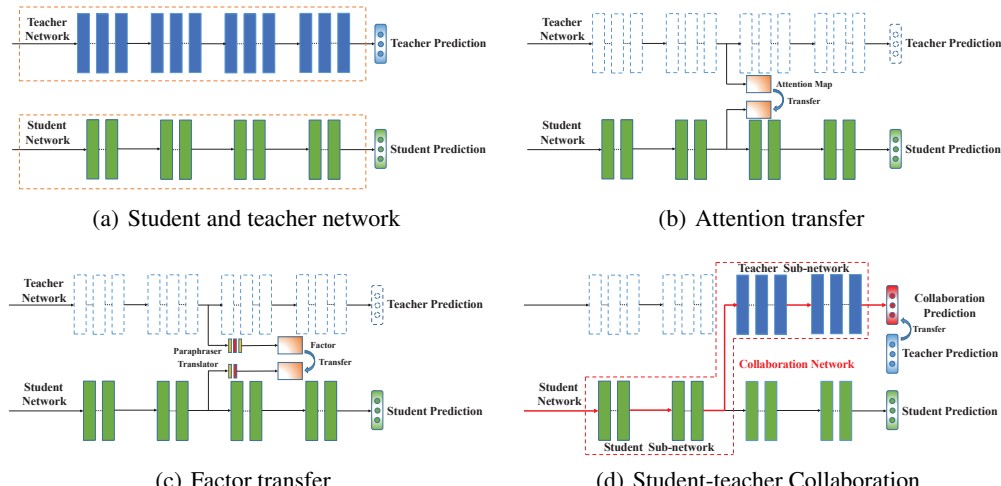

(a) Student and teacher network   (b) Attention transfer

(c) Factor transfer   (d) Student-teacher Collaboration

Figure 1: Overview of the proposed student-teacher collaboration method compared with other methods. (a) Student and teacher network. (b) Attention transfer (c) Factor transfer (d) Student-teacher collaboration. Different from the previous methods, we employ a collaboration network which is a connection of the front part of the student network and the back part of the teacher network. The difference between the predictions from the collaboration network and the teacher network is taken into account of loss during the training process. It can be clearly seen that our STC method additionally utilizes the knowledge from the top part of the teacher network.

To address the above problems, we propose a novel knowledge transfer method as illustrated in Figure 1(d). Different from the previous methods, we employ a collaboration network which is a connection of the front part of the student network and the back part of the teacher network during the training process. Specifically, we select a set of corresponding layers from the student and teacher networks. The front part and the back part of the selected layers of student and teacher networks are called student sub-network and teacher sub-network respectively. The teacher sub-network takes the intermediate representation from the student sub-network as input to make the prediction. Unlike KD using the soft target, in our method, the output tensor from the teacher network is directly treated as the target of the collaboration network. In this manner, our method can be applied to different tasks. During the training process, the difference between the predictions from the collaboration network and the teacher network is taken into account of the loss. Through back propagation, the gradient signal in the back part of the teacher network can be transferred to the student network and supervises the training process. In this way, teacher network in our method instructs student network on how to get the "answer", rather than just give an intermediate representation for mimicking as in previous methods. It can be clearly seen from Figure 1 that our method additionally utilizes the knowledge from the teacher sub-network, compared with the previous methods. It is worth noting that the collaboration network is only used during the training process, the student network with the original structure is used for prediction during the inference time.

Our contributions can be summarized as follows:

- We propose a novel knowledge transfer method, additionally utilizing the knowledge from the back part of the teacher network by employing a collaboration network structure. To the best of our knowledge, the training strategy of insturcting the student network with a collaboration network has never been used.

- The teacher network in our method instructs student network on how to get the right "answer", rather than just give an intermediate representation for mimicking as in previous methods.

- We take the output tensor from the teacher network as the target of the collaboration network, which brings good generalizability to our method on different tasks.

- We experimentally show that our method outperforms other methods with conventional strategy on various datasets in different tasks.

## 2 RELATED WORKS

To reduce the model size as well as the computational costs of deep neural networks, a variety of methods have been proposed. These methods can be summarized into five categories: network pruning, parameter quantization, tensor decomposition, efficient architecture design and knowledge transfer. Network pruning is a way to reduce the redundancy in the neural networks(LeCun et al., 1990). Han et al. (2015) pruned the network by removing the unimportant connections. Later network pruning methods operate at channel or filter levels(Han et al., 2015; Yamamoto & Maeno, 2018; Liu et al., 2018; He et al., 2017). Parameter quantization aims at compressing the model by reducing the number of bits occupied by the weights or neurons. In Gupta et al. (2015), Vanhoucke et al. (2011), Zhuang et al. and Rastegari et al. (2016), authors trained convolutional neural networks using 16-bit, 8-bit, 4-bit and 1-bit weights respectively. Han et al. (2016) optimized the model combined with network pruning, quantization and huffman coding. Tensor decomposition compresses the networks by decomposing dense convolutional kernels with low-rank approximations, including CP-decomposition(Lebedev et al., 2014), Tucker decomposition(Kim et al., 2016) and tensor ring decomposition(Zhao et al., 2016). Efficient network architecture design is an interesting approach to accelerate the model, such as SqueezeNet (Iandola et al., 2017), Mobilenet (Howard et al., 2017), ShuffleNet (Zhang et al., 2018) and Xception (Chollet, 2017).

In addition to the above four strategies that will partially change the components of the network, knowledge transfer is another way to compress the model by improving the performance of a lighter student network with the knowledge from a stronger teacher network. To the best of our knowledge, the earliest work of knowledge transfer is Bucilua et al. (2006), which to train a small model with data labeled by an ensemble of large model. Li et al. (2014) took the KL divergence between the posterior probabilities produced by the softmax operation from student and teacher model as loss function for knowledge transfer. Hinton et al. (2015) proposed a method called knowledge distillation (KD), converting the posterior probabilities extracted from teacher network into "soft targets". In Fitnets (Romero et al., 2015), they regarded the feature map extracted from the teacher network as hints, and trained the student network by mimicking the corresponding feature maps of student and teacher networks. Attention transfer (AT) (Zagoruyko & Komodakis, 2016a) computed the summations of the feature map across the channel to generate the attention map and trained the student network by minimizing the difference between the attention map of corresponding blocks. Factor transfer (FT) (Kim et al., 2018) employed a paraphraser and a translator to translate the knowledge in the feature maps into factors. The student network is optimized by minimizing the $l_2$ loss between student and teacher factors. In Ding et al. (2019), authors compressed the CNN-DBLSTM model on OCR task with Tucker decomposition and the knowledge in the teacher's BLSTM and inner product layers. There are also studies on knowledge transfer utilizing the adversarial networks, such as Xu et al., Belagiannis et al. (2018) and Wang et al. (2018).

## 3 METHOD

An eminent teacher should not only give a referenced answer to a student, but also teach student how to get the answer. Analogously, during the training process of the student network, a teacher network should teach the student network how to make the right prediction to play a role as an eminent teacher. In this section, we will firstly explain some concerns in the previous knowledge transfer methods and then describe our method in detail to explain how the teacher network instructs the student network's learning process in our method.

### 3.1 CONCERNS IN THE PREVIOUS METHODS

Previous knowledge transfer methods, such as AT (Zagoruyko & Komodakis, 2016a) and FT (Kim et al., 2018), train the student network by minimizing the difference between the intermediate representations extracted from the corresponding layers of student and teacher networks. However, this training strategy contains the following problems.

The first problem is that the teacher network in the previous methods simply provided an intermediate representation for imitation but did not teach student network how to get it. Therefore, the guidance from teacher network is limited in the previous methods. Another problem is the intermediate representation from teacher network is treated as the target in these method. However, due

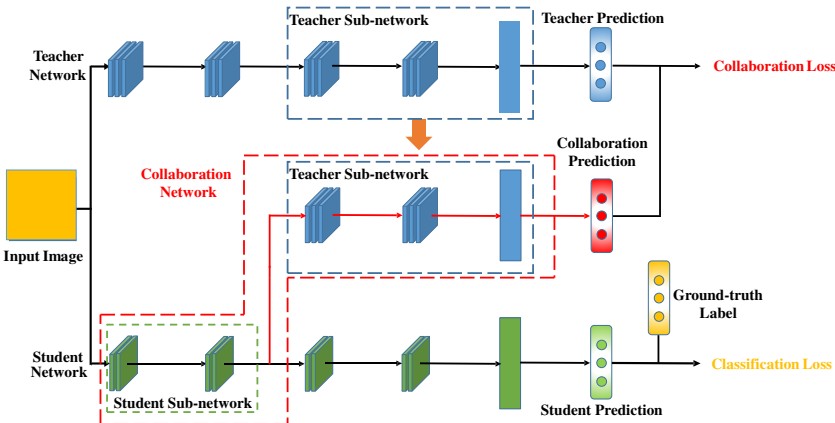

Figure 2: Training process of the proposed student-teacher collaboration method. We take the output tensor from the teacher network as the target of the collaboration network. The student network is trained with the collaboration loss and the classification loss. During the training process, the weights of the teacher network and teacher sub-network are fixed, only the weights of student network are updated.

to the divergence of the structure between the student and teacher networks, the student network usually cannot generate the same intermediate representation as the teacher network. In this case, an intermediate representation with the smallest difference does not equal to an accurate prediction. What's more, the intermediate representation is extracted from the hidden layer of the network, so the knowledge in the subsequent layers of the teacher network cannot be utilized.

## 3.2 STUDENT-TEACHER COLLABORATION

To address the above problems, we proposed a novel training strategy for knowledge transfer in our method which is insturcting the student network with a collaboration network during the training process. As illustrated in Figure 2, our method consists of three steps. Firstly, we forward propagate a pretrained teacher network $\mathcal{T}$ with the input data $\boldsymbol{I}$ to get the target $\boldsymbol{O}_t$. Secondly, we connect all the front part of a selected layer of the student network which we called student sub-network to the corresponding layer[1] of the teacher network as a collaboration network. The subsequent layers of the teacher network which we called teacher sub-network takes the intermediate representation from the student sub-network as input to produce the prediction $\boldsymbol{O}_c$. The forward propagation of the collaboration network can be formulated as:

$$\boldsymbol{O}_c = \mathcal{T}_{\boldsymbol{sub}}(\boldsymbol{x}_s^{l_s}) = \mathcal{T}_{\boldsymbol{sub}} \circ f(\boldsymbol{w}_s^{l_s} \boldsymbol{x}_s^{l_s-1}) \tag{1}$$

where $c$ and $s$ are the symbol for collaboration and student network respectively. $\mathcal{T}_{\boldsymbol{sub}}$ denotes the calculations of the teacher sub-network. $f$ denotes the activation function. $l$, $x$ and $w$ denotes the layer number, intermediate representation and weights of the network respectively. Biases are omitted for simplifying notations.

After forward propagate the collaboration network, we compute the difference between the predictions from the collaboration network and the teacher network as the STC loss $\boldsymbol{L}_{STC}$. The back propagation of STC loss can be formulated as:

$$\frac{\partial \boldsymbol{L}_{STC}(\boldsymbol{O}_c, \boldsymbol{O}_t)}{\partial \boldsymbol{w}^{l_s}} = \frac{\partial \boldsymbol{L}_{STC}(\boldsymbol{O}_c, \boldsymbol{O}_t)}{\partial \boldsymbol{O}_c} \frac{\partial \boldsymbol{O}_c}{\partial \boldsymbol{x}_s^{l_s}} \frac{\partial \boldsymbol{x}_s^{l_s}}{\partial \boldsymbol{w}_s^{l_s}} \tag{2}$$

where $\boldsymbol{O}_t$ denotes the output tensor from teacher network. Biases are omitted for simplifying notations.

---

[1]Take ResNet(He et al., 2016) as an example. The spatial size of the feature maps extracted from some layers are downsampled by 2x in ResNet architecture. In this paper, the corresponding layers of student and teacher networks are the layers that have same output spatial size after downsampling.

As can be seen in Eq. 2, the teacher network instructs the training process of the student network in a gradient signal manner. To be specific, the gradient signal of the teacher sub-network $\frac{\partial \boldsymbol{O}_c}{\partial \boldsymbol{x}_s^{l_s}}$ plays a role of weight parameters in the backward propagation formula, which indicates that the teacher network provides guidance to the student network on which element in the weights $\boldsymbol{w}^{l_s}$ should be paid more attention to during the training process. Since we directly take the output tensor of the teacher network as the target of the collaboration network, this training strategy is more accurate than minimizing the difference between the intermediate representations as previous methods did. Moreover, it can be clearly seen from Figure 2 that our method additionally utilizes the back part of the teacher network, which will bring more knowledge for student network during the training process.

There are also optional selections of target in our method, which are soft target as in Hinton et al. (2015) and the ground-truth target. However, since there is no knowledge of the teacher network in the ground-truth target, it is not a good choice for our method. As for the soft target, it can only be applied to classification task, because it is generated by the softmax function with temperature T. We will demonstrate the experimental results of different selections of the target in Sec. 4.

### 3.3 Loss function

The loss function $\boldsymbol{L}_{total}$ for training student network in classification task can be separated into two items, *i.e.* the classification loss $\boldsymbol{L}_{CF}$ and the student-teacher collaboration (STC) loss $L_{STC}$ :

$$\boldsymbol{L}_{total} = \alpha \boldsymbol{L}_{STC} + (1 - \alpha)\boldsymbol{L}_{CF} \tag{3}$$
$$\boldsymbol{L}_{STC} = \mathcal{C}(\boldsymbol{\mathcal{T}_{sub}} \circ \boldsymbol{\mathcal{S}}_{l_s}(\boldsymbol{I}) - \boldsymbol{\mathcal{T}}(\boldsymbol{I})) \tag{4}$$
$$\boldsymbol{L}_{CF} = \mathcal{C}(\boldsymbol{\mathcal{S}}(\boldsymbol{I}), \boldsymbol{G}) \tag{5}$$

where $\mathcal{C}$ denotes the cross entropy loss, $\boldsymbol{\mathcal{S}}_{l_s}$ denotes the front part calculations of the student network and $\boldsymbol{G}$ denotes the ground-truth label respectively.

We train the student network by minimizing the weighted sum of two loss as shown in Eq. 3, where weight $\alpha$ is a hyper-parameter. Specifically, the STC loss is the cross-entropy between the predictions from collaboration and teacher networks, the classification loss is the cross-entropy between the prediction from student network and ground-truth. During the training process, the weights of the teacher sub-network are fixed, only the weights of the student network are updated.

For other types of tasks, the training process can be achieved by replacing the cross-entropy loss with the corresponding loss, such as smooth-$L_1$ loss for bounding-box regression (Girshick, 2015).

### 3.4 Integrating with KD

Knowledge distillation (KD) Hinton et al. (2015) trains the student network with the soft target from teacher network. The definition of soft target is $p = softmax(\frac{o}{T})$, where $o$ is the output tensor of teacher logits (pre-softmax activations) and $T$ is a temperature. Since KD takes no additional storage and computational costs during training, it can be integrated with other knowledge transfer methods, such as AT Zagoruyko & Komodakis (2016a) and FT Kim et al. (2018) in the classification task. However, these methods result in performance degradation when integrating with KD in some cases, because of the fact that mimicking the intermediate representations does not equal to a good prediction of student network.

In contrast, our method takes the output tensor from teacher network as the target, which is consistent with KD. During experiments, our method shows good synergy integrating with KD, thus further improves the performance of student network. We will demonstrate these results in Sec. 4.

## 4 Experiments

In this section, we demonstrate the performance of proposed STC method in classification task on CIFAR-10 (Krizhevsky et al., 2014), CIFAR-100 (Krizhevsky et al., 2009), ImageNet LSVRC 2012 (Deng et al., 2009) datasets and object detection task on PASCAL VOC 2007 (Everingham & Winn, 2006) dataset. Firstly, we evaluate the STC performance integrated with or without KD (Hinton et al., 2015) of various student and teacher models on CIFAR-10, CIFAR-100 and ImageNet

Table 1: Top-1 classification error (%) on CIFAR-10 dataset. The first 7 columns are from manuscript of Kim et al. (2018).

| Student | Teacher | KD | AT | +KD | FT | +KD | STC | +KD |
|---------|---------|-----|-----|-----|-----|-----|-----|-----|
| ResNet-20 (7.78) | ResNet-56 (6.36) | 7.19 | 7.13 | 6.89 | 6.85 | 7.04 | **6.53** | **6.3** |
| ResNet-20 (7.78) | WRN-40-1 (6.52) | 7.09 | 7.34 | 7.00 | 6.85 | 6.95 | **6.30** | **6.22** |
| VGG-13 (5.99) | WRN-46-4 (4.22) | 5.71 | 5.54 | 5.30 | 4.84 | 4.65 | **4.82** | **4.63** |
| WRN-16-1 (8.62) | WRN-16-2 (5.99) | 7.64 | 8.10 | 7.52 | 7.64 | 7.59 | **7.15** | **7.05** |

Table 2: Top-1 classification error (%) on CIFAR-100 dataset. The first 7 columns are from manuscript of Kim et al. (2018).

| Student | Teacher | KD | AT | +KD | FT | +KD | STC | +KD |
|---------|---------|-----|-----|-----|-----|-----|-----|-----|
| ResNet-20 (31.24) | ResNet-110 (26.99) | 33.14 | 31.04 | 34.78 | 29.08 | 32.19 | **28.75** | **28.53** |
| ResNet-56 (28.04) | ResNet-110 (26.99) | 27.96 | 27.28 | 28.01 | 25.62 | 26.93 | **25.33** | **24.92** |

ILSVRC2012 datasets, compared with AT (Zagoruyko & Komodakis, 2016a) and FT (Kim et al., 2018). Secondly, we evaluate our method on PASCAL VOC 2007 dataset for object detection task compared with FT to show the generalizability of our method. Finally, we illustrate the result of different types of target on CIFAR-10 dataset.

## 4.1 CIFAR

CIFAR-10 and CIFAR-100 both are the basic image classification datasets and are used to evaluate the performance in many knowledge transfer methods (Zagoruyko & Komodakis, 2016a; Kim et al., 2018; Huang & Wang, 2017). For these two sets, we train the student network for 500 epochs with a mini-batch size of 128 and weight decay of $10^{-4}$. The learning rate starts from 0.1 and is divided by 10 at 150, 300 and 400 epochs. $\alpha$ is set to 0.3 on CIFAR-10 and 0.5 on CIFAR-100 empirically. For data augmentation, we following the policy in He et al. (2016). In order to make the experimental results more convincing, we use the same student and teacher networks as FT (Kim et al., 2018) did, including ResNet (He et al., 2016), Wide ResNet (WRN) (Zagoruyko & Komodakis, 2016b) and VGG (Simonyan & Zisserman, 2014). For CIFAR-10, the corresponding student and teacher networks are 1) ResNet-20 and ResNet-56, which have same width (number of channels) and different depth (number of layers). 2) ResNet-20 and WRN-40-1, which have different width and depth. 3) WRN-16-1 and WRN-16-2, which have same depth and different width. 4) VGG-13 and WRN-46-4, where residual block exists in WRN but not in VGG. For CIFAR-100, the corresponding student and teacher networks are 1) ResNet-56 and ResNet-110. 2) Resnet-20 and ResNet-110. For cases that the intermediate representation from the student sub-network has different number of channels with the teacher sub-network, a simple convolutional layer is employed to transform the dimension.

In Table 1 and Table 2, "Student" column shows the type of student network and the number in parentheses is the performance of student network trained from scratch. "Teacher" column provides the type of teacher network and the performance of pretrained teacher network by our implementation. "+KD" column provides the performance of the corresponding method combined with KD.

Performance on CIFAR-10 is shown in Table 1, for all the cases our STC method outperforms other knowledge transfer methods no matter with or without KD. We can find another result that AT and our method show good synergy when integrating with KD on CIFAR-10 dataset, while FT has conflicts with KD in some cases.

As shown in Table 2, on CIFAR-100 dataset, our method achieves the best performance among all the methods in both cases of integrated with or without KD. Different from CIFAR-10 dataset, after combining with KD, both AT and FT have an obvious performance degradation on CIFAR-100. This is because the data of CIFAR-100 are more complex than CIFAR-10 and the teacher network (ResNet-110) is deeper than that of CIFAR-10. In this case, the knowledge from the teacher sub-network is vital for the student network.

## 4.2 IMAGENET

To demonstrate the performance of our method on large dataset, we choose ResNet-18 as student network and ResNet-34 as teacher network training on ImageNet ILSVRC2012 dataset, which consists of 1.2 million training images and 50 thousand validation images. We optimize the student network with a mini-batch size of 512 and weight decay of $10^{-4}$ on 4 GPUs. The $\alpha$ are set to 0.5 empirically. The learning rate starts from 0.1 and is divided by 10 when the error plateaus. A 224×224 crop is randomly sampled from an image or its horizontal flip for data augmentation. We evaluate the performance of KD, AT, FT and proposed method STC. Top-1 and Top-5 error rates as shown in Table 3.

As can be seen, our STC method consistently outperforms other methods on both Top-1 and Top-5 error rates on ImageNet dataset. KD method suffers from the gap of depths between teacher and student network, leads to an even worse performance than training the student from scratch. KT and AT, again, show the conflicts when combined with KD, since they target at mimicking the intermediate representation but not the prediction. In contrast, our STC method improve the performance of student network by 1.36% Top-1 error rate without KD and 1.61% Top-1 error rate combined with KD, consistently shows good synergy integrated with KD.

Table 3: Top-1 and Top-5 classification error (%) on ImageNet ILSVRC2012 dataset. Pretrained student and teacher network are from PyTorch (Paszke et al., 2017) model zoo. Other statistics are gathered from our implementations.

| error (%) | Student | Teacher | KD | AT | +KD | FT | +KD | STC | +KD |
|---|---|---|---|---|---|---|---|---|---|
| Top-1 | ResNet-18 (30.24) | ResNet-34 (26.69) | 33.77 | 29.52 | 32.80 | 29.08 | 30.30 | 28.88 | 28.63 |
| Top-5 | ResNet-18 (10.92) | ResNet-34 (8.58) | 12.29 | 9.95 | 11.89 | 9.75 | 10.47 | 9.66 | 9.54 |

## 4.3 PASCAL VOC 2007

To verify the generalizability of our method on different tasks, we evaluate the performance of our method on PASCAL VOC 2007 detection dataset. We use Faster-RCNN (Ren et al., 2015) pipeline for evaluation as Kim et al. (2018) did, which are VGG-16 backbone for student network and ResNet-101 backbone for teacher network specifically. We train the student network for 15 epochs with a mini-batch size of 16. The learning rate starts from 0.01 and is divided by 10 at 6 and 11 epochs. The $\alpha$ are set to 0.5 empirically. Since the proposal boxes from RPN module in teacher and student network are of different spatial positions, we extract the RPN and detector module of the teacher network as the teacher sub-network and the backbone of student network as student sub-network, which can be seen in Figure 3.

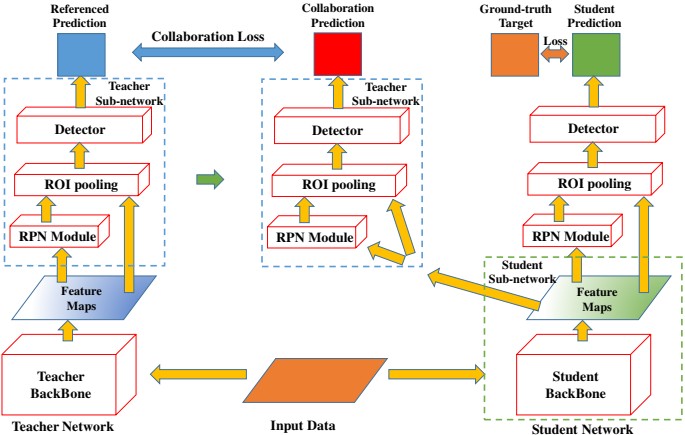

Figure 3: STC method on object detection task. We take the backbone of the student network as student sub-network and RPN, ROI pooling, detector module of teacher network as teacher sub-network.

Table 4 is the performance of our method compared with FT on PASCAL VOC 2007 dataset. As can be seen, the mAP of the student network is promoted by 1.6% in our method, while that in FT is 0.8%. This result proves that our method can be applied to various tasks and is superior to the previous methods.

Table 4: Mean average precision (%) on PASCAL VOC 2007 dataset. Both student and teacher network follow the Faster-RCNN pipeline. The backbone of student and teacher networks are VGG-16 and ResNet-101 respectively.

| Method | Student | Teacher | mAP |
|--------|---------|---------|-----|
| FT | VGG-16 backbone(69.4) | ResNet-101 backbone (75.2) | 70.3 |
| STC | VGG-16 backbone(69.4) | ResNet-101 backbone (75.2) | **71.0** |

## 4.4 ABLATION STUDY

In this paragraph, we evaluate the performance of different selections of the target in our method on CIFAR-10 dataset, using the same student and teacher network as Sec. 4.1.

The performance of various target can be seen in Table 5. In Table 5, "output tensor (t)" and "soft target (t)" denote the output tensor and the soft target are generated from the teacher networks. "ground-truth (d)" denotes the ground-truth target is from the dataset. For the generating of the soft target, we follow the fashion in Hinton et al. (2015).

As can be seen, the selection of ground-truth target gets the worst results among all the cases. This is because the ground-truth target does not contain knowledge in the teacher network, so there is limited help in improving the performance of the student network.

For the selection of soft target, even though it can get about the same performance, it can only be employed on classification task, because the soft target is produced by the softmax function with temperature T. Considering the generalizability of our method on different tasks, we choose the output tensor from teacher network as the target.

Table 5: Top-1 classification error(%) of different selections of the target on CIFAR-10 dataset.

| Student | Teacher | output tensor (t) | soft target (t) | ground-truth (d) |
|---------|---------|-------------------|-----------------|------------------|
| ResNet-20 (7.78) | ResNet-56 (6.39) | **6.53** | 6.59 | 6.78 |
| ResNet-20 (7.78) | WRN-40-1 (6.84) | **6.30** | 6.55 | 6.83 |
| VGG-13 (5.99) | WRN-46-4 (4.44) | 4.82 | **4.80** | 5.17 |
| WRN-16-1 (8.62) | WRN-16-2 (6.27) | **7.15** | 7.39 | 8.01 |

## 5 CONCLUSION

In this paper, we proposed a novel knowledge transfer method called student-teacher collaboration (STC). Different from previous methods, our method employs a collaboration network by connecting the front part of the student network and the back part of the teacher network during the training process. We take the difference between the output predictions from the collaboration network and the teacher network into account of the loss to train the student network. Through back propagation, the knowledge of the teacher sub-network can be additionally utilized in a gradient signal manner. Specifically, the teacher network provides guidance to the student network on which element in the weights should be paid more attention to during the training process.

Through plentiful experiments, it is proved that our STC method outperforms previous knowledge transfer methods on various datasets and has good generalizability on different tasks. What's more, our method has good synergy integrated with KD to further improve the performance of the student network, while other methods result in accuracy degradation in some cases.

To the best of our knowledge, the training strategy which is insturcting the student network with a collaboration network has never been used in other knowledge transfer methods. We believe that this novel idea will further promote the development of knowledge transfer.

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
