# OpenReview forum: "Knowledge Transfer via Student-Teacher Collaboration"
_ICLR.cc/2020/Conference — Reject_

### Official Review · AnonReviewer3 · 2019-10-16
**Official Blind Review #3**

**Rating:** 8

**Review:**

The paper suggests a new method for knowledge transfer from teacher neural network to student: student-teacher collaboration (STC). The idea is that teacher not only provides student with the correct answer or correct intermediate representation, but also instructs student network on how to get the right answer. The paper suggests to merge the front part of the student network and back part of the teacher network into the collaboration network. In this way, the weights in the student subnetwork are learned from the loss backpropagated from the teacher subnetwork. The target labels for collaboration network are outputs of teacher network. The method is adapted for different tasks, classification and bounding box regression are presented in the experiments. It outperforms previous methods on various datasets. Furthermore, the method shows good results when integrated with traditional knowledge distillation.

Overall, the paper is a significant algorithmic contribution. The related work section provides thorough review of different methods to decrease computational costs, including not only knowledge distillation, but also pruning, compressing and decomposition approaches. The idea is elegant and, to the best of my knowledge, has never been suggested in other works. Considering the theoretical part, it is clearly shown how the gradient signal from the teacher sub-network guides the student network on which part of the weights should be paid attention on. All derivations are easy to follow. The paper also considers how the suggested idea is aimed to solve the problems of previous knowledge transfer methods. The experimental section is consistent and clearly shows the advantage of the suggested method. Teacher and student networks used are different sizes of ResNet, Wide ResNet and VGG. The paper presents classification experiments on CIFAR-10, CIFAR-100, ImageNet and object detection experiment on PASCAL VOC 2007 dataset. STC outperforms previous methods, both with KD integration and without. The performance is always better than pure student training (which was not always the case for previous methods) and sometimes the results are even better than teacher performance. Finally, the choice of teacher output as target over soft target and ground truth, which was previously motivated in the theoretical section, is shown to be superior in the experiment.

Possible improvement of the paper is the instruction on how to choose the intermediate layer from where to teach the representation, i.e. where the student sub-network ends and teacher sub-network begins. For object detection experiment the choice of the border is naturally defined by the architecture of the network in Faster-RCNN approach. Could the choice be different? May be somewhere inside the BackBone part of the networks? For classification, it could be interesting to study how this choice influences the results. However, this question didn’t affect my score of the paper, and, as far as I know, it is also not considered in the previous works on knowledge distillation.

Minor comments
1.	In the context of accelerating the models using decomposition, Lebedev et al., ICLR 2015 could be cited.
2.	Page 2: difference tasks -> different tasks
3.	Page 2 the first bullet point: additionally utilizes -> additionally utilizing/which additionally utilizes
4.	Page 2 the third bullet point: brings good generalizability -> which brings good generalizability
5.	Page 5: “training strategy is more accelerate than” – odd phrase
6.	Page 6: while KT has conflicts with KD in some cases -> while FT has conflicts with KD in some cases.


**Experience Assessment:**

I have published one or two papers in this area.

**Review Assessment: Checking Correctness Of Derivations And Theory:**

I carefully checked the derivations and theory.

**Review Assessment: Checking Correctness Of Experiments:**

I assessed the sensibility of the experiments.

**Review Assessment: Thoroughness In Paper Reading:**

I read the paper thoroughly.

---

> ### Author Response · Authors · 2019-11-14
> **Responses to Review #3**
>
> Dear Reviewer #3:
>
> We would like to extend our sincere thanks to you for your positive comments and constructive feedback. We also want to thank you for taking the time to patiently check the grammar and expression errors in our paper. We have corrected all the typos based on your suggestions in the updated version of our paper.
>
> For the improvement of our paper which is about the choice of the intermediate layer selections, we believe this is a very meaningful work. An instruction on how to choose the intermediate layer from where to teach the representation allows student network to improve performance more effectively during knowledge transfer process. Thank you for your constructive suggestions and we will study it in our future work.
>
> Best regards,

---

### Official Review · AnonReviewer1 · 2019-10-18
**Official Blind Review #1**

**Rating:** 6

**Review:**


This paper proposed a new method for knowledge distillation, which transfers knowledge from a large teacher network to a small student network in training to help network compression and acceleration. The proposed method concatenate the first a few layers of the student network with the last a few layers of the teacher network, and claims the gradient directly flows from teacher to student, instead of through a KL or L2 similarity loss between teacher and student logits.

The experimental results look good, and extensive experiments have been done on CIFAR, ImageNet and PASCAL VOC.

However, the description of the proposed method looks rather unclear. First, the ‘front’ and ‘back’ part of networks are very vague. I have to guess that is the first a few layers of student and last a few layers of teacher. And it is still unclear how many layers in student and teacher are concatenated to form the ‘collaboration network’. How could the authors connect the two subnetwork with different structures?

It is unclear to me why proposed method is better than AT, FT or FitNets. It looks to me the proposed method use an ad-hoc selected layer to transfer knowledge from teacher to student, and the transfer is indirect because it has to go through the pre-trained subnetwork in teacher.

Minor issue, FT and AT are not defined when they first appear in page 1.

Could the authors show the student and teacher accurayunder standard supervised training in the result tables?

Several related works are not discussed, such as
Xu et al. 2018 https://arxiv.org/abs/1709.00513
Belagiannis et al. 2018 https://arxiv.org/abs/1803.10750
Wang et al. 2018 https://papers.nips.cc/paper/7358-kdgan-knowledge-distillation-with-generative-adversarial-networks


============ after rebuttal ================
I updated my rate to weak accept. Though it is a borderline or below paper to me. The paper has really good empirical results. However, I cannot understand the intuition behind the paper why concatenating teacher to student is better than use l2 for intermediate layers. The choice of the transferring layer seems to be rather ad-hoc, and it is hard to say how much tuning needed to get the empirical benefits.

**Experience Assessment:**

I have published one or two papers in this area.

**Review Assessment: Checking Correctness Of Derivations And Theory:**

N/A

**Review Assessment: Checking Correctness Of Experiments:**

I assessed the sensibility of the experiments.

**Review Assessment: Thoroughness In Paper Reading:**

I made a quick assessment of this paper.

---

> ### Author Response · Authors · 2019-11-14
> **Responses to Review #1**
>
> Dear reviewer #1:
>
> We would like to extend our sincere thanks to you for your constructive feedback.
> We feel sorry for your confusion. In our method, the student sub-network is all the front part of a selected layer of the student network and teacher sub-network is all the subsequent layers of teacher network corresponding to the selected layer. The definition of the corresponding layers is mentioned in the updated version of our paper and the description of our proposed method have been modified, which can be seen on page 4.  For cases that the intermediate representation from the student sub-network has different number of channels with the teacher sub-network, a simple convolutional layer is employed to transform the dimension. We have pointed this out in the first paragraph of Sec. 4.1. in the updated version of our paper.
>
> For your other concerns, here are our responses:
>
> Q: It is unclear to me why proposed method is better than AT, FT or FitNets. It looks to me the proposed method use an ad-hoc selected layer to transfer knowledge from teacher to student, and the transfer is indirect because it has to go through the pre-trained subnetwork in teacher.
>
> A: Our theory is described in Sec. 3.2. The gradient signal of the teacher sub-network plays a role of weight parameters in the backward propagation formula, which indicates that the teacher network provides guidance to the student network on which element in the weights of student network should be paid more attention to during the training process. For the previous methods, they transfer the knowledge by minimizing the loss between the intermediate representations from student and teacher network. However, due to the divergence of the structure between the student and teacher networks, the student network cannot generate the total same intermediate representation as the teacher network. In this case, these methods are not able to determine which element in the weights of student network need to be paid more attention to during the training process.
>
> Q: Minor issue, FT and AT are not defined when they first appear in page 1.
>
> A: The corresponding illustrations of AT and FT are mentioned in page 1. In order to reduce the redundancy of the article, we describe the AT and FT methods briefly in page 1 and define these two methods in page 2.
>
> Q: Could the authors show the student and teacher accuracy under standard supervised training in the result tables?
>
> A: For all the experiment results in our paper, the value in the parentheses after the network’s type is the
> accuracy of student and teacher network under standard supervised training.
>
> Q: Several related works are not discussed
>
> A: The related works you mentioned have been included in our updated version. Due to the page limitation mentioned in ICLR submission instructions, these work can only be described briefly in our article. We hope to get your understanding.
>
> Best regards,

---

### Official Review · AnonReviewer2 · 2019-10-23
**Official Blind Review #2**

**Rating:** 3

**Review:**

Overall the method proposed in this paper is simple but effective, and adequate experimental results are given to show its performance improvements.  However, the literature survey of this paper is not satisfactory.

1. To reduce model size, there are several different ways including efficient architecture design, parameter pruning, quantization, tensor decomposition and knowledge distillation. The authors forgot to mention tensor decomposition and mixed it with efficient architecture design. As for parameter pruning and quantization,  many important papers are missing.

2. Utilizing the "soft targets" to transfer knowledge from teacher to student model is not first proposed by Hinton et al. (2015). To the best of my knowledge, it is first proposed in
J. Li, R. Zhao, J.-T. Huang, Y. Gong, “Learning small-size DNN with output-distribution-based criteria,” Proc. Interspeech-2014, pp.1910-1914.

3. Leveraging back part of teacher model's guidance to improve student performance has been investigated by other researchers on OCR tasks in
Ding H, Chen K, Huo Q. Compressing CNN-DBLSTM models for OCR with teacher-student learning and Tucker decomposition[J]. Pattern Recognition, 2019, 96: 106957.
They combine student's CNN with teacher's DBLSTM to learn better representations.

In conclusion, I will give a weak reject currently, unless the authors improve their literature survey and modify their claims.





**Experience Assessment:**

I have published one or two papers in this area.

**Review Assessment: Checking Correctness Of Derivations And Theory:**

I assessed the sensibility of the derivations and theory.

**Review Assessment: Checking Correctness Of Experiments:**

I assessed the sensibility of the experiments.

**Review Assessment: Thoroughness In Paper Reading:**

I read the paper thoroughly.

---

> ### Author Response · Authors · 2019-11-14
> **Responses to Review #2**
>
> Dear reviewer #2,
>
> We would like to extend our sincere thanks to you for your constructive feedback. We have modified our claims and improved our literature survey based on your comments. Here are our responses to your major concerns.
>
> Q: To reduce model size, there are several different ways including efficient architecture design, parameter pruning, quantization, tensor decomposition and knowledge distillation. The authors forgot to mention tensor decomposition and mixed it with efficient architecture design. As for parameter pruning and quantization, many important papers are missing.
>
> A: We apologize for the unclear description in our “Related Work” section and have improved the literature survey based on your suggestions in the updated version of our paper. Due to the page limitation mentioned in ICLR submission instructions, some of the related works can only be described briefly in our article. We hope to get your understanding.
>
> Q: Utilizing the "soft targets" to transfer knowledge from teacher to student model is not first proposed by Hinton et al. (2015). To the best of my knowledge, it is first proposed in J. Li, R. Zhao, J.-T. Huang, Y. Gong, “Learning small-size DNN with output-distribution-based criteria,” Proc. Interspeech-2014, pp.1910-1914.
>
> A: In paper “Learning small-size DNN with output-distribution-based criteria”, authors took the KL divergence between the posterior probabilities produced by the softmax operation from student and teacher model as loss function for knowledge transfer. Hinton et al. (2015) improved the posterior probability from teacher network by employing the softmax function on the teacher logits with temperature T and called it as “soft target”. The definition of “soft target” in our paper is the same as in Hinton’s. In other knowledge transfer methods, such as FitNet, AT and FT, the definition of “soft target” is the same as ours. We have cited the prior paper in the updated version.
>
> Q: Leveraging back part of teacher model's guidance to improve student performance has been investigated by other researchers on OCR tasks in Ding H, Chen K, Huo Q. Compressing CNN-DBLSTM models for OCR with teacher-student learning and Tucker decomposition[J]. Pattern Recognition, 2019, 96: 106957. They combine student's CNN with teacher's DBLSTM to learn better representations.
>
> A: We are sorry for missing this paper in our literature survey. However, our work started half a year ago, while this paper is published on July 7, 2019. We have cited this paper in the updated version of our paper.
> In paper “Compressing CNN-DBLSTM models for OCR with teacher-student learning and Tucker decomposition”, authors employed a knowledge distillation method with DarkNet-DBLSTM as student network and VGG-DBLSTM as teacher network. The DBLSTM modules of the student and teacher networks in this paper have same topology, so the student’s BLSTM and inner product layers can borrow parameters from the teacher’s counterparts during training and inference. In contrast, the student networks in our proposed method do not take any part of teacher network for inference and the back part of the teacher network is only employed during the training process. Therefore, our method can be generalized to situations where the student network and the teacher network have different structure. Besides, our method can also be applied to different tasks, which has been confirmed through our experiments.
>
>
> Best regards,

---

### Public Comment · ~Grigory_V._Sapunov1 · 2019-10-13
**Unclear details of comparison**

You mentioned, "KD method suffers from the gap of depths between teacher and student network, leads to an even worse performance than training the student from scratch".

As I understand, you took pretrained teacher and student networks from PyTorch model zoo, in this sense these networks are heavily optimized and well trained.

Then you performed KD, but it's unclear, how does your procedure (initialization, size of datasets, length of training and so on) compare to that one used for pretrained models.

Is it possible that KD works worse just because you trained the models on less data or for a shorter time? Are your methods comparable?

---

> ### Author Response · Authors · 2019-10-14
> **Reply**
>
> Hi,
> Thank you for your comment! For the experiments on ImageNet dataset, we employ the same hyper-parameters as ResNet (He et al., 2016) for all methods and all data from ImageNet dataset are fully used. It can be seen from the  manuscript of Factor Transfer (Kim et al., 2018), the conclusion of KD method on ImageNet dataset is same as ours.

---

### Author Response · Authors · 2019-11-14
**Summary of updates in the paper**

We would like to extend our sincere thanks to three reviewers for their constructive feedback and helpful comments. We have made the following major modifications to our manuscript:

1.	We have improved our literature survey. More related works are discussed in our paper on page 3.

2.	We have modified our claims on contributions and conclusion on page 2 and 8.

3.	The definition of the corresponding layers is mentioned in the footnote on page 4.

4.	“For cases that the intermediate representation from the student sub-network has different number of channels with the teacher sub-network, a simple conv layer is employed to transform the dimension.”  This is pointed out in the first paragraph of Sec. 4.1.

5.	We have corrected all the typos in the updated version of our paper.

---

### Public Comment · ~Erika_Taylor1 · 2020-08-08
**Interdisciplinary addition?**

I don't know if it's just me, but I think this paper would benefit from a small excursus adding a sociocultural element to it. As I've read on https://www.myhubintranet.com/knowledge-transfer/, any kind of KT bridging two generations (which is absolutely the case between students and teachers) comes along with certain specifics and characteristics.

---

### Decision · Program_Chairs · 2019-12-19

**Decision:**

Reject

**Comment:**

This paper has been assessed by three reviewers scoring it as follows: 6, 3, 8. The submission however attracted some criticism post-rebuttal from the reviewers e.g., why concatenating teacher to student is better than the use l2 loss or how the choice of transf. layers has been made (ad-hoc). Similarly, other major criticism includes lack of proper referencing to parts of work that have been in fact developed earlier in preceding papers. On balance, this paper falls short of the expectations of ICLR 2020, thus it cannot be accepted at this time. The authors are encouraged to work through major comments and resolve them for a future submission.